# MOMENTUM CONTRASTIVE AUTOENCODER

## ABSTRACT

Wasserstein autoencoder (WAE) shows that matching two distributions is equivalent to minimizing a simple autoencoder (AE) loss under the constraint that the latent space of this AE matches a pre-specified prior distribution. This latent space distribution matching is a core component in WAE, and is in itself a challenging task. In this paper, we propose to use the contrastive learning framework that has been shown to be effective for self-supervised representation learning, as a means to resolve this problem. We do so by exploiting the fact that contrastive learning objectives optimize the latent space distribution to be uniform over the unit hyper-sphere, which can be easily sampled from. This results in a simple and scalable algorithm that avoids many of the optimization challenges of existing generative models, while retaining the advantage of efficient sampling. Quantitatively, we show that our algorithm achieves a new state-of-the-art FID of 54.36 on CIFAR-10, and performs competitively with existing models on CelebA in terms of FID score. We also show qualitative results on CelebA-HQ in addition to these datasets, confirming that our algorithm can generate realistic images at multiple resolutions.

## 1 INTRODUCTION

The main goal of generative modeling is to learn a given data distribution while facilitating an efficient way to draw samples from them. Popular algorithms such as variational autoencoders (VAE, Kingma & Welling (2013)) and generative adversarial networks (GAN, Goodfellow et al. (2014)) are theoretically-grounded models designed to meet this goal. However, they come with some challenges. For instance, VAEs suffer from the posterior collapse problem (Chen et al., 2016; Zhao et al., 2017; Van Den Oord et al., 2017), and a mismatch between the posterior and prior distribution (Kingma et al., 2016; Tomczak & Welling, 2018; Dai & Wipf, 2019; Bauer & Mnih, 2019). GANs are known to have the mode collapse problem (Che et al., 2016; Dumoulin et al., 2016; Donahue et al., 2016) and optimization instability (Arjovsky & Bottou, 2017) due to their saddle point problem formulation.

With the Wasserstein autoencoder (WAE), Tolstikhin et al. (2017) propose a general theoretical framework that can potentially avoid these challenges. They show that the divergence between two distributions is equivalent to the minimum reconstruction error, under the constraint that the marginal distribution of the latent space is identical to a prior distribution. The core challenge of this framework is to match the latent space distribution to a prior distribution that is easy to sample from. If this challenge is addressed appropriately, WAE can avoid many of the aforementioned challenges of VAE and GANs. Tolstikhin et al. (2017) investigate GANs and maximum mean discrepancy (MMD, Gretton et al. (2012)) for this task and empirically find that the GAN-based approach yields better performance despite its instability. Others have proposed solutions to overcome this challenge (Kolouri et al., 2018; Knop et al., 2018), but they come with their own pitfalls (see Section 2).

This paper aims to design a generative model that avoids the aforementioned challenges of existing approaches. To do so, we build on the WAE framework. In order to tackle the latent space distribution matching problem, we make a simple observation that allows us to use the contrastive learning framework to solve this problem. Contrastive learning achieves state-of-the-art results in self-supervised representation learning tasks (He et al., 2020; Chen et al., 2020) by forcing the latent representations to be 1) augmentation invariant; 2) distinct for different data samples. It has been shown that the contrastive learning objective corresponding to the latter goal pushes the learned representations to achieve maximum entropy over the unit hyper-sphere (Wang & Isola, 2020). We observe that applying this contrastive loss term to the latent representation of an AE therefore matches it to the uniform distribution over the unit hyper-sphere. This approach avoids the aforementioned

optimization challenges of existing methods, thus resulting in a simple and scalable algorithm for generative modeling that we call Momentum Contrastive Autoencoder (MoCA).

## 2 RELATED WORK

There are many autoencoder based generative models in existing literature. One of the earliest model in this category is the de-noising autoencoder (Vincent et al., 2008). Bengio et al. (2013b) show that training an autoencoder to de-noise a corrupted input leads to the learning of a Markov chain whose stationary distribution is the original data distribution it is trained on. However, this results in inefficient sampling and mode mixing problems (Bengio et al., 2013b; Alain & Bengio, 2014).

Variational autoencoders (VAE) (Kingma & Welling, 2013) overcome these challenges by maximizing a variational lower bound of the data likelihood, which involves a KL term minimizing the divergence between the latent's posterior distribution and a prior distribution. This allows for efficient approximate likelihood estimation as well as posterior inference through ancestral sampling once the model is trained. Despite these advantages, followup works have identified a few important drawbacks of VAEs. The poor sample qualities of VAE has been attributed to a mismatch between the prior (which is used for drawing samples) and the posterior (Kingma et al., 2016; Tomczak & Welling, 2018; Dai & Wipf, 2019; Bauer & Mnih, 2019). The VAE objective is also at the risk of posterior collapse – learning a latent space distribution which is independent of the input distribution if the KL term dominates the reconstruction term (Chen et al., 2016; Zhao et al., 2017; Van Den Oord et al., 2017).

Dai & Wipf (2019) claim that the reason behind poor sample quality of VAEs is a mismatch between the prior and posterior, arising from the latent space dimension of the autoencoder being different from the intrinsic dimensionality of the data manifold (which is typically unknown). To overcome this mismatch, they propose to learn a two stage VAE in which the second stage learns a VAE on the latent space samples of the first. They show that this two stage training and sampling significantly improves the quality of generated samples. However, training a second VAE is computationally expensive and introduces some of the same challenges mentioned above.

Ghosh et al. (2019) observe that VAEs can be interpreted as deterministic autoencoders with noise injected in the latent space as a form of regularization. Based on this observation, they introduce deterministic autoencoders and empirically investigate various other regularizations. The further introduce a post-hoc density estimation for the latent space since the autoencoding step does not match it to a prior. In this context, one can view our proposed algorithm as a way to regularize deterministic autoencoders while simultaneously learning a latent space distribution which can be easily sampled from.

Tolstikhin et al. (2017) make the observation that the optimal transport problem can be equivalently framed as an autoencoder objective (WAE) under the constraint that the latent space distribution matches a prior distribution. They experiment with two alternatives to satisfy this constraint in the form of a penalty – MMD (Gretton et al., 2012) and GAN (Goodfellow et al., 2014)) loss, and they find that the latter works better in practice. Training an autoencoder with an adversarial loss was also proposed earlier in adversarial autoencoders (Makhzani et al., 2015). Our algorithm builds on the aforementioned WAE theoretical framework due to its theoretical advantages.

There has been research that aims at avoiding the latent space distribution matching problem all together by making use of sliced distances. For instance, Kolouri et al. (2018) observe that Wasserstein distance for one dimensional distributions have a closed form solution. Motivated by this, they propose to use sliced-Wasserstein distance, which involves a large number of projections of the high dimensional distribution onto one dimensional spaces which allows approximating the original Wasserstein distance with the average of one dimensional Wasserstein distances. A similar idea using the sliced-Cramer distance is introduced in Knop et al. (2018). However, the number of required random projections becomes prohibitively high when the data lives on a low dimensional manifold in a high dimensional space, making this approach computationally inefficient or otherwise inaccurate (Liutkus et al., 2019).

## 3 MOMENTUM CONTRASTIVE AUTOENCODER

We present the proposed algorithm in this section. We begin by restating the WAE theorem that connects the autoencoder loss with the Wasserstein distance between two distributions. Let $X \sim P_X$

be a random variable sampled from the real data distribution on $\mathcal{X}$, $Z \sim Q(Z|X)$ be its latent representation in $\mathcal{Z} \subseteq \mathbb{R}^d$, and $\hat{X} = g(Z)$ be its reconstruction by a deterministic decoder/generator $g : \mathcal{Z} \to \mathcal{X}$. Note that the encoder $Q(Z|X)$ can also be deterministic in the WAE framework, and we let $f(X) \stackrel{dist}{=} Q(Z|X)$ for some deterministic $f : \mathcal{X} \to \mathcal{Z}$.

**Theorem 1.** *(Bousquet et al., 2017; Tolstikhin et al., 2017) Let $P_Z$ be a prior distribution on $\mathcal{Z}$, let $P_g = g\#P_Z$ be the push-forward of $P_Z$ under $g$ (i.e. the distribution of $\hat{X} = g(Z)$ when $g \sim P_Z$), and let $Q_Z = f\#P_X$ be the push-forward of $P_X$ under $f$. Then,*

$$W_c(P_X, P_g) = \inf_{Q:Q_Z=P_Z} \mathbb{E}_{\substack{X \sim P_X \\ Z \sim Q(Z|X)}} [c(X, g(Z))] = \inf_{f:f\#P_X=P_Z} \mathbb{E}_{X \sim P_X} [c(X, g(f(X)))] \quad (1)$$

*where $W_c$ denotes the Wasserstein distance for some measurable cost function c.*

The above theorem states that the Wasserstein distance between the true ($P_X$) and generated ($P_g$) data distributions can be equivalently computed by finding the minimum (w.r.t. $f$) reconstruction loss, under the constraint that the marginal distribution of the latent variable $Q_Z$ matches the prior distribution $P_Z$. Thus the Wasserstein distance itself can be minimized by jointly minimizing the reconstruction loss w.r.t. both $f$ (encoder) and $g$ (decoder/generator) as long as the constraint is met.

In this work, we parameterize the encoder network $f : \mathcal{X} \to \mathbb{R}^d$ such that latent variable $Z = f(X)$ has unit $\ell_2$ norm. Our goal is then to match the distribution of this $Z$ to the uniform distribution over the unit hyper-sphere $\mathcal{S}_d = \{z \in \mathbb{R}^d : \|z\|_2 = 1\}$. To do so, we study the so-called "negative sampling" component of the contrastive loss used in self-supervised learning,

$$L_{neg}(f;\tau, K) = \mathbb{E}_{\substack{x \sim P_X \\ \{x_i^-\}_{i=1}^K \sim P_X}} \left[ \log \frac{1}{K} \sum_{j=1}^{K} e^{f(x)^T f(x_j^-)/\tau} \right] \quad (2)$$

Here, $f : \mathcal{X} \to \mathcal{S}_d$ is a neural network whose output has unit $\ell_2$ norm, $\tau$ is the temperature hyper-parameter, and $K$ is the number of samples (another hyper-parameter). Theorem 1 of Wang & Isola (2020) shows that for any fixed $t$, when $K \to \infty$,

$$\lim_{K \to \infty} (L_{neg}(f;\tau, K) - \log K) = \mathbb{E}_{x \sim P_X} \left[ \log \mathbb{E}_{x^- \sim P_X} \left[ e^{f(x)^T f(x^-)/\tau} \right] \right] \quad (3)$$

Crucially, this limit is minimized *exactly* when the push-forward $f\#P_X$ (i.e. the distribution of the latent random variable $Z = f(X)$ when $X \sim P_X$) is uniform on $\mathcal{S}_d$. Moreover, even the Monte Carlo approximation of Eq. 2 (with mini-batch size $B$ and some $K$ such that $B \leq K < \infty$)

$$L_{neg}^{MC}(f;\tau, K, B) = \frac{1}{B} \sum_{i=1}^{B} \log \frac{1}{K} \sum_{j=1}^{K} e^{f(x_i)^T f(x_j)/\tau} \quad (4)$$

is a consistent estimator (up to a constant) of the entropy of $f\#P_X$ called the redistribution estimate (Ahmad & Lin, 1976). This follows if we notice that $k(x_i;\tau, K) := \sum_{j=1}^{K} e^{f(x_i)^T f(x_j)/\tau}$ is the un-normalized kernel density estimate of $f(x_i)$ using the i.i.d. samples $\{x_j\}_{j=1}^K$, so $-L_{neg}^{MC}(f;\tau, K, B) = -\frac{1}{B} \sum_{i=1}^{B} \log k(x_i;\tau, K)$ (Wang & Isola, 2020). So minimizing $L_{neg}$ (and importantly $L_{neg}^{MC}$) maximizes the entropy of $f\#P_X$.

Tolstikhin et al. (2017) attempted to enforce the constraint that $f\#P_X$ and $P_Z$ were matching distributions by regularizing the reconstruction loss with the MMD or a GAN-based estimate of the divergence between $f\#P_X$ and $P_Z$. By letting $P_Z$ be the uniform distribution over the unit hyper-sphere $\mathcal{S}_d$, the insights above allow us to instead minimize the much simpler regularized loss

$$L(f, g; \lambda, \tau, B, K) = \frac{1}{B} \sum_{i=1}^{B} \|x_i - g(f(x_i))\|_2^2 + \lambda L_{neg}^{MC}(f;\tau, K, B) \quad (5)$$

**Training**: For simplicity, we will now use the notation $\text{Enc}(\cdot)$ and $\text{Dec}(\cdot)$ to respectively denote the encoder and decoder network of the autoencoder. Further, the $d$-dimensional output of $\text{Enc}(\cdot)$ is $\ell_2$ normalized, i.e., $\|\text{Enc}(x)\|_2 = 1 \, \forall x$. Based on the theory above, we aim to minimize the

---

**Algorithm 1** PyTorch-like pseudocode of Momentum Contrastive Autoencoder algorithm

---

```
# Enc_q, Enc_k: encoder networks for query and key. Their outputs are L2 normalized
# Dec: decoder network
# Q: dictionary as a queue of K randomly initialized keys (dxK)
# m: momentum
# lambda: regularization coefficient for entropy maximization
# tau: logit temperature

for x in data_loader: # load a minibatch x with B samples
    z_q = Enc_q(x) # queries: Bxd
    z_k = Enc_k(x).detach() # keys: Bxd, no gradient through keys
    x_rec = Dec(z_q) # reconstructed input

    # positive logits: Bx1
    l_pos = bmm(z_q.view(B,1,d), z_k.view(B,d,1))

    # negative logits: BxK
    l_neg = mm(z_q.view(B,d), Q.view(d,K))

    # logits: Bx(1+K)
    logits = cat([l_pos, l_neg], dim=1)

    # compute loss
    labels = zeros(B) # positive elements are in the 0-th index
    L_con = CrossEntropyLoss(logits/tau, labels) # contrastive loss maximizing entropy of z_q
    L_rec = ((x_rec - x) ** 2).sum() / B # reconstruction loss
    L = L_rec + lambda * L_con # momentum contrastive autoencoder loss

    # update Enc_q and Dec networks
    L.backward()
    update(Enc_q.params)
    update(Dec.params)

    # update Enc_k
    Enc_k.params = m * Enc_k.params + (1-m) * Enc_q.params

    # update dictionary
    enqueue(Q, z_k) # enqueue the current minibatch
    dequeue(Q) # dequeue the earliest minibatch
```

---

bmm: batch matrix multiplication; mm: matrix multiplication; cat: concatenation.

enqueue appends $Q$ with the keys $z_k \in \mathbb{R}^{B \times d}$ from the current batch; dequeue removes the oldest $B$ keys from $Q$

---

loss $L(\text{Enc}, \text{Dec}; \lambda, \tau, B, K)$, where $\lambda$ is the regularization coefficient, $\tau$ is the temperature hyperparameter, $B$ is the mini-batch size, and $K \geq B$ is the number of samples used to estimate $L_{neg}$.

In practice, we propose to use the momentum contrast (MoCo, He et al. (2020)) framework to implement $L_{neg}$. Let $\text{Enc}_t$ be parameterized by $\theta_t$ at step $t$ of training. Then, we let $\text{Enc}_t'$ be the same encoder parameterized by the exponential moving average $\tilde{\theta}_t = (1-m) \sum_{i=1}^{t} m^{t-i} \theta_i$. Letting $x_1, \ldots, x_K$ be the $K$ most recent training examples, and letting $t(j) = t - \lfloor j/B \rfloor$ be the time at which $x_j$ appeared in a training mini-batch, we replace $L_{neg}^{MC}$ at time step $t$ with

$$L_{MoCo} = \frac{1}{B} \sum_{i=1}^{B} \log \frac{1}{K} \sum_{j=1}^{K} \exp\left(\frac{\text{Enc}_t(x_i)^T \text{Enc}_{t(j)}'(x_j)}{\tau}\right) - \frac{1}{B} \sum_{i=1}^{B} \frac{\text{Enc}_t(x_i)^T \text{Enc}_t'(x_i)}{\tau} \quad (6)$$

This approach allows us to use the latent vectors of inputs outside the current mini-batch without re-computing them, offering substantial computational advantages over other contrastive learning frameworks such as SimCLR (Chen et al., 2020). Forcing the parameters of $\text{Enc}'$ to evolve according to an exponential moving average is necessary for training stability, as is the second term encouraging the similarity of $\text{Enc}_t(x_i)$ and $\text{Enc}_t'(x_i)$ (so-called "positive samples" in the terminology of contrastive learning). Note that we do not use any data augmentations in our algorithm, but this similarity term is still non-trivial since the networks $\text{Enc}_t$ and $\text{Enc}_t'$ are not identical. Pseudo-code of our final algorithm, which we call Momentum Contrastive Autoencoder (MoCA), is shown in Algorithm 1 (pseudo-code style adapted from He et al. (2020)). Finally, in all our experiments, inspired by Grill et al. (2020) we set the exponential moving average parameter $m$ for updating the $\text{Enc}'$ network at the $t^{th}$ iteration as $m = 1 - (1 - m_0) \cdot (\cos(\pi t/T) + 1)/2$, where $T$ is the total number of training iterations, and $m_0$ is the base momentum hyper-parameter.

**Inference**: Once the model is trained, the marginal distribution of the latent space (i.e. the pushforward $\text{Enc} \# P_X$) should be close to a uniform distribution over the unit hyper-sphere. We can therefore draw samples from the learned distribution as follows: we first sample $z \sim \mathcal{N}(0, I)$ from the standard multivariate normal distribution in $\mathbb{R}^d$ and then generate a sample $x_g := \text{Dec}(z/\|z\|_2)$.

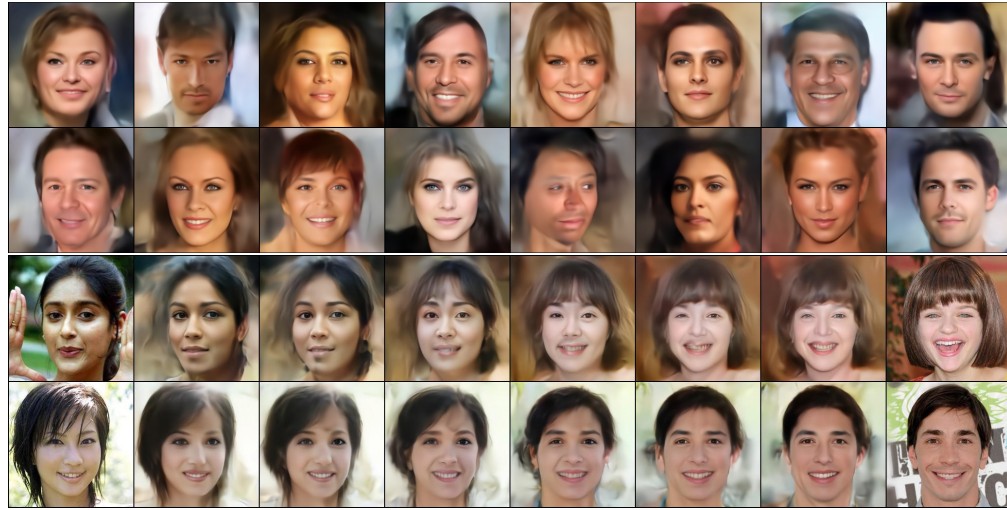

Figure 1: Random samples (rows 1-2) from a model trained with MoCA on CelebA-HQ, and that model's interpolations (rows 3-4) between images in latent space. The leftmost and rightmost columns of rows 3-4 are the original images from the test set of CelebA-HQ which we are interpolating.

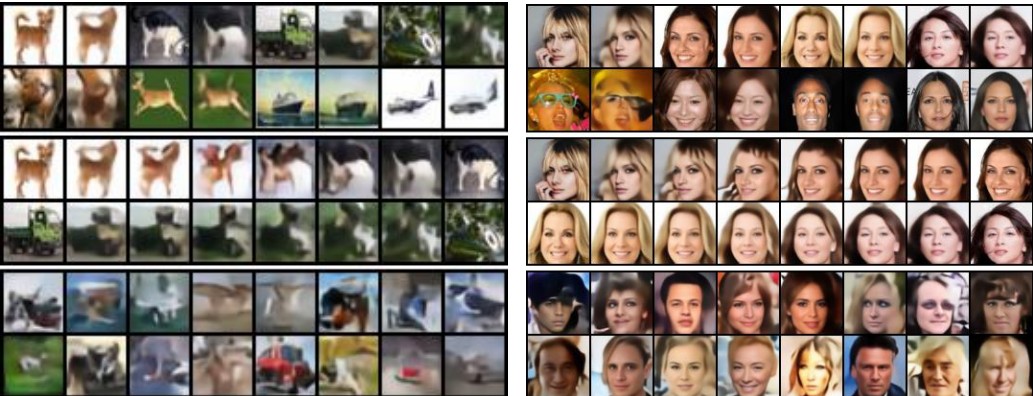

Figure 2: **Left**: CIFAR-10. **Right**: CelebA. Rows 1-2 show original image (odd column) and its reconstruction (even column). Rows 3-4 show model's interpolation between two test images in latent space. The leftmost and rightmost columns of rows 3-4 are the original images from the corresponding test set. Rows 5-6 show random samples drawn from a trained model.

## 4 EXPERIMENTS

We present a quantitative and qualitative evaluation of the samples generated by Momentum Contrastive Networks trained on CelebA (Liu et al., 2015), CIFAR-10 (Krizhevsky et al., 2009), and CelebA-HQ (Karras et al., 2018). For all datasets except CelebA-HQ, we use two architectures: A1: the architecture from Tolstikhin et al. (2017), which is commonly used as a means to fairly compare against existing methods; and A2: a ResNet-18 based architecture with much fewer parameters. For CelebA-HQ, we use a variant of ResNet-18 with 6 residual blocks instead of 4 for both the encoder and decoder. The remaining details are provided in Appendix A.

For quantitative analysis we report the Fréchet Inception Distance (FID) score (Heusel et al., 2017). In Table 1, we compare the performance of MoCA with VAE (Kingma & Welling, 2013), WAE (Tolstikhin et al., 2017), two stage VAE (Dai & Wipf, 2019), and spectral normalized regularized autoencoder (RAE-SN, Ghosh et al. (2019)). The numbers for 2-Stage VAE and RAE-SN are cited from their respective papers. The CelebA numbers for VAE are WAE are cited from Tolstikhin et al. (2017), while those for CIFAR-10 are cited from Dai & Wipf (2019). We achieve a new state-of-the-art on CIFAR-10 using our model with the A2 architecture, and we achieve competitive performance on CelebA.

Qualitatively, we visualize random (not cherry-picked) samples from our trained models on all the datasets. Figure 1 (rows 1-2) contains random samples from CelebA-HQ. Figure 2 (rows 5-6) contains

| FID\Model | VAE | WAE-MMD | WAE-GAN | 2-Stage VAE | RAE-SN | MoCA-A1 | MoCA-A2 |
|-----------|-----|---------|---------|-------------|--------|---------|---------|
| CelebA | 63 | 55 | 42 | **34** | 40.95 | 48.43 | 44.59 |
| CIFAR-10 | 106 | 80.9 | - | 72.9 | 75.30 | 77.49 | **54.36** |

Table 1: Evaluation of MoCA against existing baselines using FID (lower is better, best models in bold). MoCA-A1 uses an architecture similar to the one used in WAE (Tolstikhin et al., 2017), while MoCA-A2 uses a ResNet-18 based architecture.

random samples from CIFAR-10 and CelebA, as well as reconstructions (rows 1-2) of images from these datasets. Most generated samples look realistic, even across multiple resolutions.

We also present latent space interpolations between images from the test set of CelebA-HQ in Figure 1 (rows 3-4). We present latent space interpolations for CIFAR-10 and CelebA in Figure 2 (rows 3-4). For these interpolations, we compute the latent vectors $z = \text{Enc}(x)$ and $z' = \text{Enc}(x')$ for two images $x$ and $x'$, let $z_\alpha = \alpha z + (1 - \alpha)z'$ for some $0 \leq \alpha \leq 1$, and then generate the interpolated image $\hat{x}_\alpha = \text{Dec}\left(z_\alpha/\|z_\alpha\|_2\right)$. These latent space interpolations show that our algorithm causes the generator to learn a smooth function from the unit hyper-sphere $\mathcal{S}_d$ to image space, and moreover, almost all intermediate samples look quite realistic.

We present additional random samples, image reconstructions, and latent space interpolations for these three datasets in Appendix B.

## 5 ABLATION ANALYSIS

Unlike most existing autoencoder based generative models, our proposal of using the contrastive learning framework, specifically momentum contrastive learning (He et al., 2020) due to its computational efficiency compared to its competitor Chen et al. (2020), introduces a number of hyper-parameters in addition to the regularization coefficient $\lambda$. Therefore, it is important to shed light on their behavior during the training process of MoCA. This section explores how these various hyper-parameters impact the quality of generated samples. We also study the impact of the latent dimension of the autoencoder under MoCA training. To keep the analysis tractable and quantitative, we use the Fréchet Inception Distance (FID) score to evaluate the performance of the trained models..

For this section, unless specified otherwise, we use the CelebA dataset with the ResNet-18 autoencoder architecture (architecture A2 in the previous section), $\tau = 1$, $m_0 = 0.999$, $d = 128$, $\lambda = 3000$, $K = 60000$. For optimization, we use Adam with learning rate 0.001, batch size 64, drop this learning rate by half every 60 epochs, and train for a total of 200 epochs. All other optimization hyper-parameters are set to the default Pytorch values.

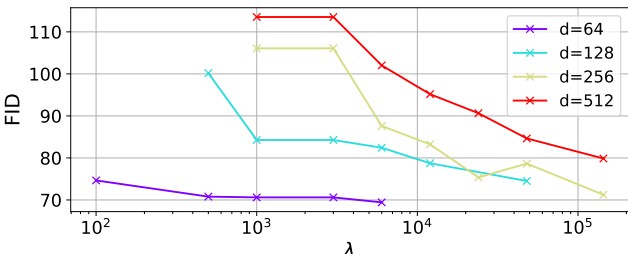

Figure 3: The interplay between the latent dimension $d$ of the MoCA network and the regularization coefficient $\lambda$. Larger $d$ requires significantly larger $\lambda$ to achieve comparable FID scores for the generated sample quality. See text for more details.

### 5.1 LATENT SPACE DIMENSIONALITY $d$ AND REGULARIZATION COEFFICIENT $\lambda$

Real data often lies on a low dimensional ($d_0 < n$) manifold that is embedded in a high dimensional ($n$) space (Bengio et al., 2013a). Autoencoders attempt to map the probability distribution of the data to a designated prior distribution in a latent space of of dimension $d$, and vice versa. However, if there is a significant mismatch in the dimension $d_0$ of the true data manifold and the latent space's dimension $d$, learning a mapping between the two becomes impossible (Dai & Wipf, 2019). This results in many "holes" in the learned latent space which do not correspond to the training data distribution.

| size | $32 \times 32$ | $64 \times 64$ | $256 \times 256$ |
|------|------|------|------|
| $\lambda^\star$ | 100 | 2000 | 20000 |

Table 2: The optimal value of $\lambda$ scales linearly with input size. We consider $\lambda$ between 100 and 50000 and report the value $\lambda^\star$ that achieves the best FID score for each input size. See Appendix C for the full data.

| $m_0$ | 0 | 0.9 | 0.99 | 0.999 |
|------|------|------|------|------|
| FID | 86.57 | 98.97 | 47.96 | 47.46 |

Table 3: Smaller base momentum $m_0$ causes model performance to degrade significantly. Performance is measured using the FID score (lower is better). Note that we use the cosine schedule described in section 3 in all these experiments.

Given the importance of this problem, we study how the latent dimension $d$ influences the quality of samples generated by our used by our model MoCA. We also simultaneously analyze the influence of the regularization coefficient $\lambda$, since the value of $\lambda$ enforces how much we want the mapped latent distribution to be close to the uniform distribution on the unit hyper-sphere.

For this experiment, we use $d \in \{64, 128, 256, 512\}$ and study the value of $\lambda$ on a wide range on the log scale between 100 and 144,000 (in some cases). Due to the large number of experiments in this analysis, we train each configuration until the epoch reconstruction loss (mean squared error) reaches 50. For this reason the FID scores are much higher than the fully trained models reported in other experiments (where reconstruction loss reaches $\sim$25). The results are shown in Figure 3. We find that for $d = 64$ the performance is quite stable across different $\lambda$ values. However, as $d$ is set to larger values, we find that a significantly larger value of $\lambda$ is required to reach a similar FID score. We hypothesize that this is because a larger $\lambda$ forces the "meaningful regions" to be more uniformly distributed in the latent space. Therefore, even though there are "holes" in the latent space (due to $d_0 < d$), a random sample from uniform distribution is more likely to be close to one of these meaningful regions.

## 5.2 Choosing $\lambda$ given input size

An important consideration when selecting the regularization coefficient $\lambda$ is the relative scale of the reconstruction loss $\|x_i - g(f(x_i))\|_2^2$ and contrastive loss. We downsample CelebA-HQ to $32 \times 32$, $64 \times 64$, and $256 \times 256$ for this experiment, and we report the value of $\lambda \in \{100, 200, 500, 1000, 2000, 5000, 10000, 20000, 50000\}$ that achieves the best FID score for each image size. We find that the optimal value of $\lambda$ is roughly proportional to the number of pixels in the input (Table 2). See Appendix C for a more detailed discussion.

## 5.3 Importance of momentum $m$

We study the impact of the contrastive learning hyper-parameter $m$ on the generated sample quality. $m$ is the exponential moving average hyper-parameter used for updating the parameters of the momentum encoder network $\text{Enc}_k$. $m$ is typically kept to be close to 1 for training stability (He et al., 2020). We confirm this intuition for our generative model as well in Table 3. We use the base value $m_0 \in \{0, 0.9, 0.999\}$. Note that we use the cosine schedule to compute the value of $m$ every iteration (as discussed in section 3), which makes $m$ increase from the base value $m_0$ to 1 over the course of training. We find that FID scores are much worse when $m$ is not close to 1.

## 5.4 Importance of temperature $\tau$

Based on the discussion below Eq. 3, the negative term in the contrastive loss essentially estimates the entropy of the latent space distribution due to its equivalent kernel density estimation (KDE) interpretation (Eq. 4). Therefore, the temperature hyper-parameter $\tau$ used in the contrastive loss acts as the smoothing parameter of this KDE and controls the granularity of the estimated distribution. Thus for larger temperature, the estimated distribution becomes smoother and the entropy estimation becomes poor, which should result in poor quality of generated samples. Additionally, for larger $\tau$, intuitively, a larger $\lambda$ should be needed in order to push the KDE samples apart from one another. We confirm these intuitions in Figure 4. Note that due to the large number of experiments in this analysis, we train each configuration until 50 epochs (explaining the inferior FID values).

## 5.5 Effect of $K$

Since we use the momentum contrastive framework, it would be useful to understand how the dictionary size $K$ affects the quality of generative model learned. The dictionary $Q$ contains the

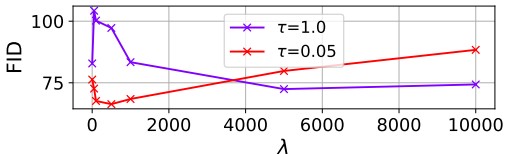 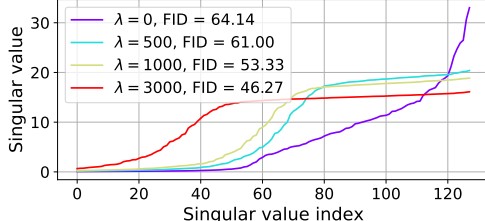

Figure 4: Impact of $\tau$ on optimal choice of $\lambda$ and best FID for generated samples. Optimal $\lambda$ is lower for lower $\tau$. Best FID is better for lower $\tau$. This suggests that entropy is maximized more accurately when $\tau$ is smaller.

Figure 5: SVD of latent representation $\in \mathbb{R}^{128}$ for models trained with various values of $\lambda$. Larger $\lambda$ results in more uniform singular values, i.e., closer to uniform distribution, and lower (better) FID for generated samples.

| $K$ | 100 | 5000 | 10,000 | 30,000 | 60,000 | 120,000 |
|---|---|---|---|---|---|---|
| FID | 84.93 | 48.45 | 46.56 | 47.31 | 46.68 | 49.94 |

Table 4: The effect of dictionary size $K$ on the quality of samples measured using the FID score (lower is better). The performance is largely stable across different values of $K$ unless $K$ is too small.

negative samples in the contrastive framework which are used to push the latent representations away from one another, encouraging the latent space to be more uniformly distributed. We therefore expect a small $K$ would be bad for achieving this goal. Our experiments in Table 4 confirm this intuition. We use $K \in \{100, 5000, 10000, 30000, 60000, 120000\}$. We find that the FID score is stable and small across the various values of $K$ chosen, except for $K = 100$, for which FID is much worse.

## 5.6 MISMATCH BETWEEN $Q_Z$ AND $P_Z$

Finally, we now try to evaluate how well the contrastive term in our objective addresses the problem of matching the marginal distribution $Q_Z = \text{Enc}\#P_X$ of the autoencoder latent space to the prior distribution $P_Z$, viz., the uniform distribution on the unit hyper-sphere. Since the encoder is parameterized to output unit $\ell_2$ norm vectors, we only need to evaluate how close $Q(Z)$ is to being isotropic. As a computationally efficient proxy, we compute the singular value decomposition (SVD) of the latent representation corresponding to 10,000 randomly sampled images from the training set. We do this for models trained with different values of the regularization coefficient $\lambda \in \{0, 500, 1000, 3000\}$. Larger $\lambda$ is designed to increase the entropy of the latent space to better match it to the uniform distribution. As Figure 5 shows, for models trained with larger $\lambda$, the singular values (and therefore the latent distribution) become more uniform. Corresponding FID scores on generated samples reflect this effect since models trained with larger $\lambda$ have lower (better) FID scores.

## 6 CONCLUSION AND FUTURE WORK

We propose a novel algorithm for learning a generative model called Momentum Contrastive Autoencoders (MoCA). The main idea behind MoCA is to use the contrastive learning framework to match the autoencoder's latent space marginal distribution with the uniform distribution on the unit hyper-sphere. Our objective has theoretical connections with Wasserstein autoencoder, but our algorithm avoids many of the optimization challenges of existing autoencoder-based generative models. We demonstrate that our algorithm can generate samples that are competitive with or better than existing state-of-the-art algorithms. Finally, we present a thorough analysis of the various hyper-parameters in our algorithm introduced due to the contrastive learning framework, and how they impact learning.

We note that contrastive learning currently yields state-of-the-art performance in self-supervised learning tasks. Since we use it as a part of our learning algorithm, a natural question is whether we can jointly perform representation learning and generative modeling using a single objective. We leave this research direction as a future work.

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

APPPENDIX

# A   TRAINING AND EVALUATION DETAILS

We ran all our experiments in Pytorch 1.5.0 (Paszke et al., 2019).

**Datasets**:

**CIFAR-10** contains $32 \times 32$ images of 50k training samples and 10k test samples.

**CelebA** dataset contains a total of $\sim$203k $64 \times 64$ images of divided into $\sim$180k training images and $\sim$20,000 test images. The images were pre-processed by first taking 140x140 center crops and then resizing to 64x64 resolution.

**CelebA-HQ** contains a total of $\sim$30k $1024 \times 1024$ images. We resized these images to $256 \times 256$ and split the dataset into $\sim$27k training images and $\sim$3k test images.

**Architecture and optimization**:

The network architecture A1 is identical to the CNN architecture used in Tolstikhin et al. (2017) except that we use batch norm (Ioffe & Szegedy, 2015) in every layer (similar to Ghosh et al. (2019)), and the latent dimension is 128 for the CelebA dataset. This architecture roughly has around 38 million parameters. We found that using 64 dimensions for CelebA in this architecture prevented the reconstruction loss from reaching small values.

The encoder of the network architecture A2 is a modification of the standard ResNet-18 architecture He et al. (2016) in that the first convolutional layer has filters of size $3 \times 3$, and the final fully connected layer has latent dimension 128. The decoder architecture is a mirrored version of the encoder with upsampling instead of downsampling. Additionally, the final convolutional layer uses an upscaling factor of 1 for CIFAR-10 and 2 for CelebA. The architecture roughly has around 24 million parameters.

Both A1 and A2 were trained on CIFAR-10 with MoCA hyperparameters $K = 30000, \tau = 0.05, m_0 = 0.99$. A1 used $\lambda = 1000$ while A2 used $\lambda = 100$. Both models were trained for 100 epochs using the Adam optimizer with batch size 64, and learning rate 0.001 decayed by a factor of 2 every 60 epochs.

Both A1 and A2 were trained on CelebA with MoCA hyperparameters $K = 60000, \tau = 0.05, m_0 = 0.99$. A1 used $\lambda = 1000$ while A2 used $\lambda = 100$. Both models were trained for 200 epochs using the Adam optimizer with batch size 64, and learning rate 0.001 decayed by a factor of 2 every 60 epochs.

During our experiments, we found that the choice of hyper-parameters $\tau$ and $m$ was stable across the two architectures and datasets and they were chosen based on our ablation studies. The value of $K$ was decided based on the size of the dataset (CelebA being larger than CIFAR-10 in our case). Finally, we found that the value of $\lambda$ was generally subjective to the dataset and architecture being used. We typically ran a grid search over $\lambda \in \{100, 1000, 3000, 6000\}$.

The images in Figure 1 (CelebA-HQ $256 \times 256$) were generated using a variant of the ResNet-18 architecture. The base ResNet-18 architecture has 4 residual blocks, each containing 2 convolutional layers and an additional convolutional layer which spatially downsamples its input by a factor of $2 \times 2$. For the encoder, we use the same architecture, but with 6 blocks (to downsample a $256 \times 256$ image to $4 \times 4$, which we then flatten and project into the latent space). The decoder is a mirrored version of the encoder, but with de-convolution upsampling layers instead of downsampling layers. The latent space of this architecture is 128 dimensional. We train this model with MoCA hyperparameters $\lambda = 20000, K = 30000, \tau = 1, m_0 = 0.99$. The model was trained for 1000 epochs using the Adam optimizer with batch size 64, and learning rate 0.002 (decayed by a factor of 2 every 40 epochs until epoch 400).

**Quantitative evaluation**:

In all our experiments, FID was always computed using the test set of the corresponding dataset. We always use 10,000 samples for computing FID.

## B  ADDITIONAL QUALITATIVE RESULTS

Figures 6, 7, and 8 respectively depict additional randomly sampled images, reconstructions, and latent space interpolations for CelebA-HQ $256 \times 256$. Figures 7 and 8 are generated by the same model used to generate Figure 1, while Figure 6 is generated by an earlier checkpoint of that model (selected for best visual quality).

Figures 9, 11, and 13 respectively depict additional randomly sampled images, reconstructions, and latent space interpolations for CIFAR-10 using the same model that achieved the FID score of 54.36 in table 1.

Figures 10, 12, and 14 respectively depict additional randomly sampled images, reconstructions, and latent space interpolations for CelebA using the same model that achieved the FID score of 44.59 in table 1.

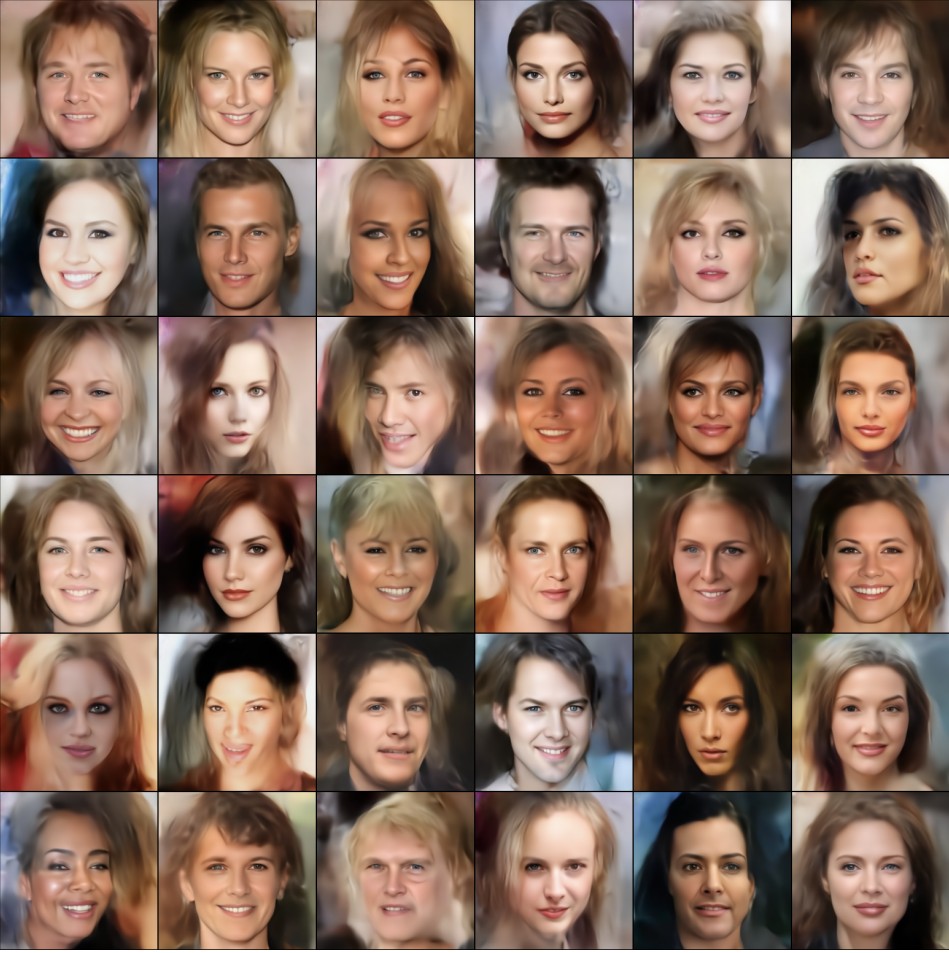

Figure 6: Random samples generated by a model trained (as described in Appendix A on CelebA-HQ $256 \times 256$ for 850 epochs. Model checkpoint picked based on best visual quality.

## C  ADDITIONAL DATA ON CHOOSING $\lambda$ BASED ON INPUT SIZE

In Section 5.2, we show that the optimal value of the regularization weight $\lambda$ scales linearly with input size. For this experiment, we downscale CelebA-HQ ($256 \times 256$) to $64 \times 64$ and $32 \times 32$, and we study the impact of $\lambda$ for the different input sizes. We construct the models for generating $d \times d$ images by removing $\log_2(256/d)$ of the 6 residual blocks from the encoder/decoder of the base model used for CelebA-HQ ($256 \times 256$) (described in Appendix A), as each block downsamples/upsamples

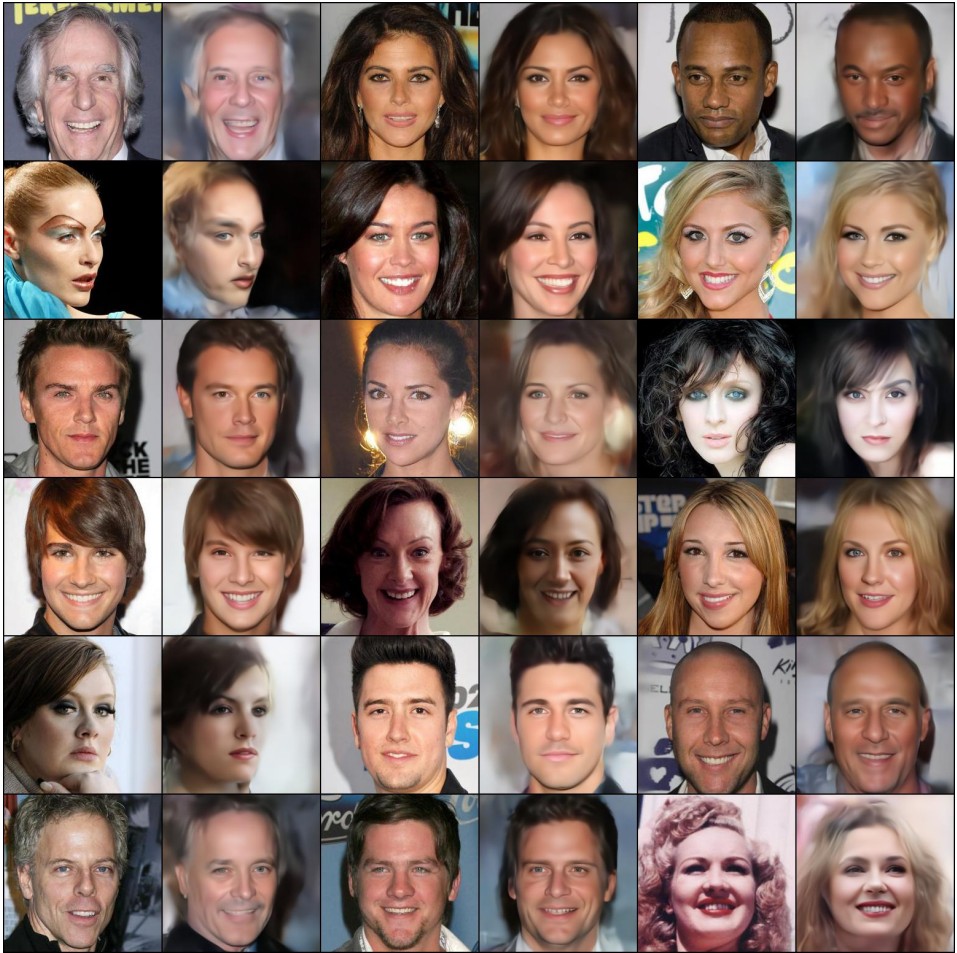

Figure 7: Image reconstructions by a model trained (as described in Appendix A) on CelebA-HQ $256 \times 256$. For each pair of columns, the left is the original image, and the right is the reconstruction.

the image by a factor of $2 \times 2$. We train all models for 400 epochs using the Adam optimizer with batch size 64 and learning rate $0.002$ (decayed by a factor of 2 every 40 epochs).

The full data for this experiment (which support the claims of Table 2) can be found in Figure 15. We consider $\lambda \in \{100, 200, 500, 1000, 2000, 5000, 10000, 20000, 50000\}$ for each image size (except $256 \times 256$, as quality rapidly deteriorates when $\lambda < 5000$). Please note that the absolute FID scores are not comparable between different image sizes! Rather, we emphasize the relative trends.

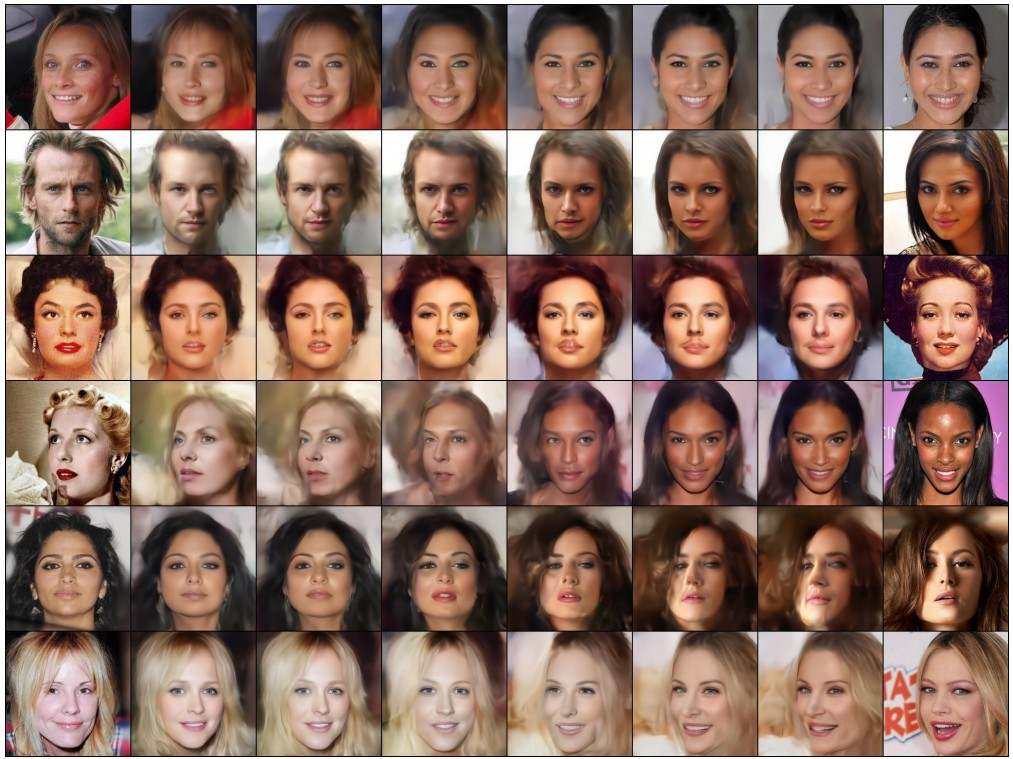

Figure 8: Latent space interpolations by a model trained (as described in Appendix A) on CelebA-HQ $256 \times 256$.

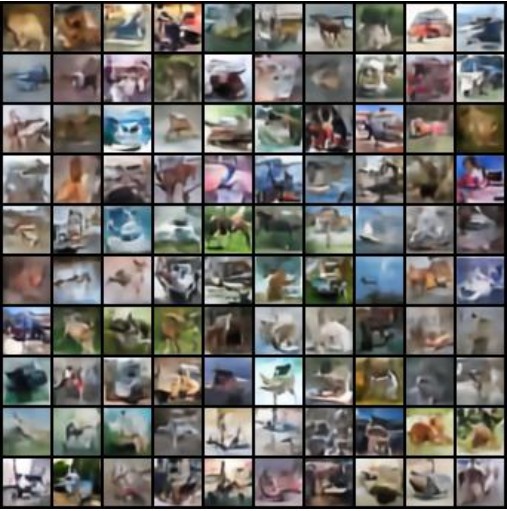

Figure 9: Randomly generated samples from the MoCA model trained on CIFAR-10 with FID 54.36 in table 1.

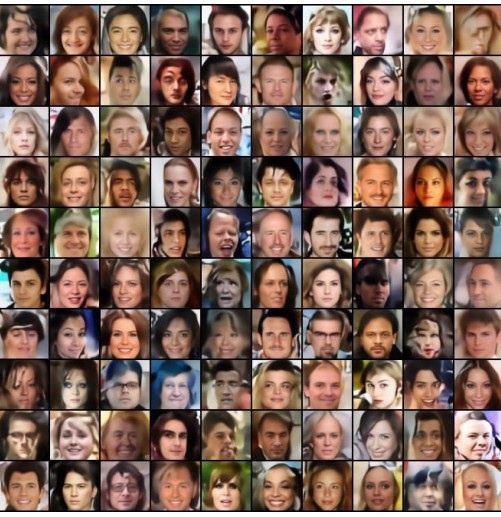

Figure 10: Randomly generated samples from the MoCA model trained on CelebA with FID 44.59 in table 1.

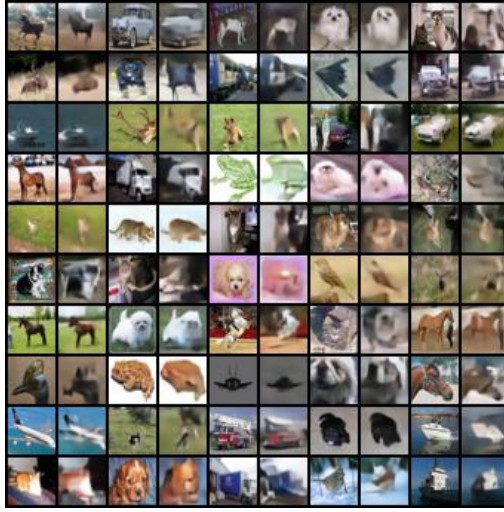

Figure 11: Reconstructed test samples from the MoCA model trained on CIFAR-10 with FID 54.36 in table 1.

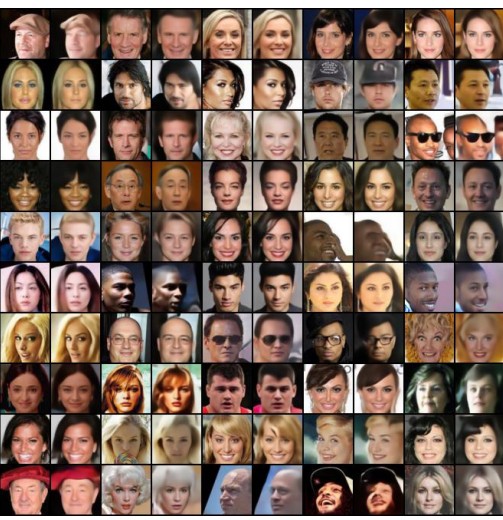

Figure 12: Reconstructed test samples from the MoCA model trained on CelebA with FID 44.59 in table 1.

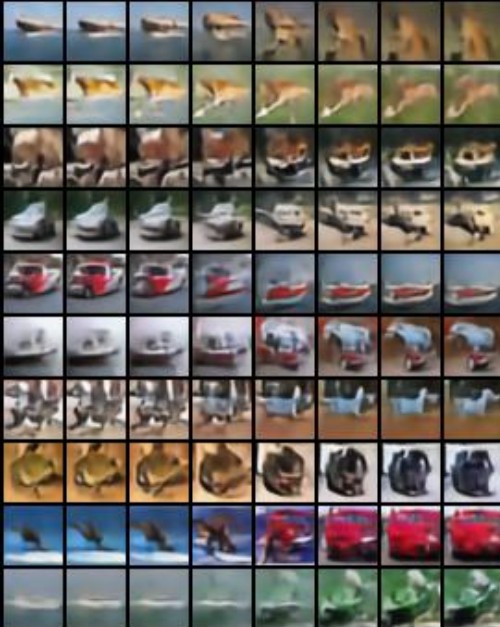

Figure 13: Interpolation between two test images in latent space for MoCA model trained on CIFAR-10 with FID 54.36 in table 1. The leftmost and rightmost columns are the original images from the corresponding test set.

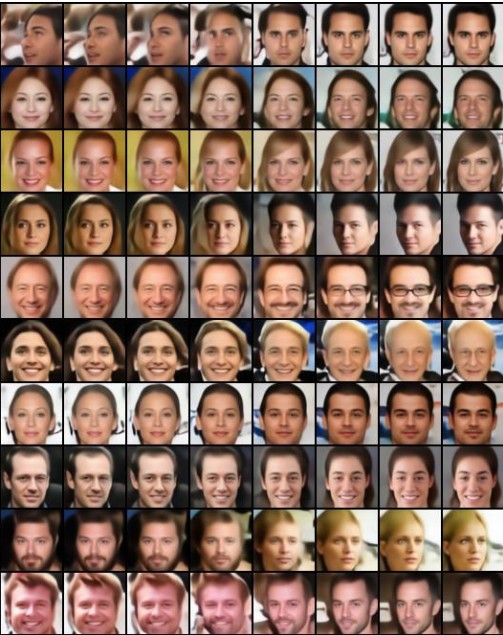

Figure 14: Interpolation between two test images in latent space for MoCA model trained on CelebA with FID 44.59 in table 1. The leftmost and rightmost columns are the original images from the corresponding test set.

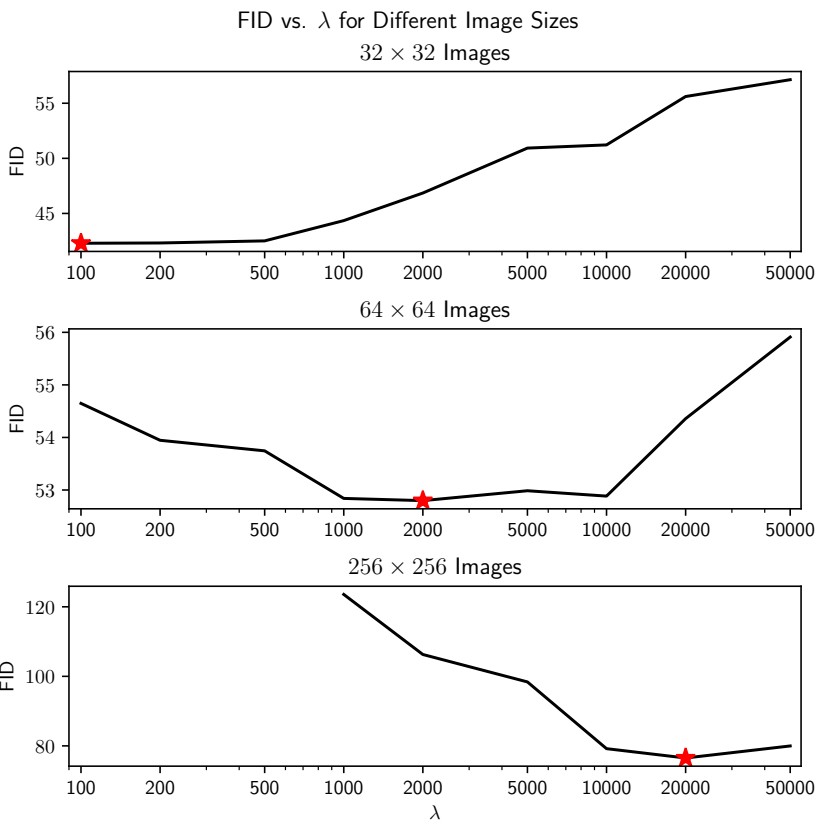

Figure 15: Impact of regularization weight $\lambda$ on FID score based on input size. Optimal values $\lambda^\star$ (labeled with a red star) scale linearly with input size. Note that absolute FID scores are not comparable between different image sizes! This figure focuses on relative trends.

