# OpenReview forum: "Momentum Contrastive Autoencoder"
_ICLR.cc/2021/Conference — Reject_

### Official Review · AnonReviewer1 · 2020-10-19
**Another generative autoencoder**

**Rating:** 4
**Confidence:** 4

**Review:**


The authors aim at constructing another generative autoencoder: reconstruction loss plus matching-aggregate-posterior-with-prior penalty. For this they use a constrastive loss in latent space.

Without paying attention to writing style and clarity, etc., the paper should be rejected to be published at ICLR already for the following reasons:

- lack of originality: any paper that follows the above recipe of minimizing reconstruction loss plus a penalty that matches (aggregate) posterior with prior needs more content than just swapping the divergence in latent space. Since contrastive losses are well known their invention can't be contributed to this paper.
This brings us to the following point:
- lack of theoretical guarantees: There are no new theoretical guarantees or insights in this paper. Empirically lowering the FID score on cifar by a few points without any guarantees is imho not good enough anymore for research in generative autoencoders.
Furthermore:
- missing citations: the authors didn't keep up with the literature. among many others they miss e.g. [PB].

[PB] G Patrini, R van den Berg, et al.: Sinkhorn AutoEncoders. UAI 2019.

---

### Official Review · AnonReviewer4 · 2020-10-26
**The main contribution is the nice theoretical connection between WAE training and contrastive learning. However, I am concerned with the statistical significance of some of the results.**

**Rating:** 4
**Confidence:** 4

**Review:**

Description: The authors propose momentum contrastive Wasserstein autoencoders (MoCA), which is an extension of the Wasserstein autoencoder (WAE) that aims to match the prior p(z) and aggregate variational posterior q(z) through the use of contrastive learning, as opposed to earlier proposed techniques (MMD, GAN). The use of contrastive learning here is theoretically motivated by the fact that -- as shown in Wang & Isola (2020) -- in the limit of infinitely many negative samples, the distribution induced by the contrastive encoder is uniform on the hypersphere. Therefore, if p(z) is the unit + uniform hypersphere, then we can leverage contrastive learning to drive the aggregate variational posterior q(z) to be as close to it as possible. This provides a principled way to train WAEs since its corresponding optimisation assumes that P_Z == Q_Z.

Advantages:
- A theoretical connection is made between contrastive learning and the training of WAEs.
- Ablations explore the intricacies of combining MoCA with WAE.
- Paper is written clearly.

Disadvantages:
- The empirical results in Table 1 suggest that MoCA is at least competitive with CelebA, though (1)  these should have uncertainty estimates and (2) the WAE-GAN result for CIFAR10 is missing. You should include uncertainty estimates over multiple runs. This may also involve you having to run WAE-GAN / WAE-MMD yourself, since I assume that you were not able to quote uncertainty estimates from their paper. Even if uncertainty estimates were in the cited papers, I *strongly recommend* to run these experiments yourself to remove any confounding variables from the fact that these numbers were quoted from the literature (since the experimental setup could be vastly different compared to yours). I'm not suggesting you do this for all the methods in Table 1, but I highly suggest you do so for the WAE experiments. Otherwise I have little confidence in the results you have presented.

Comments / suggestions / typos:

- Theorem 1, "...when g ~ P_z". Don't you mean "...when z ~ P_z"?
- Use subfigures for Figure 2 to make the presentation more clear (distinguishing between reconstructions, interpolations, and samples from the prior)
- For the FID calculation in Table 1, what is the reference dataset here? (I see in the supplementary material you say it's the test set, ok, maybe mention this in the caption for Table 1 since it's important)
- Generally, VAE papers also report the ELBO on the test set to examine generalisation performance. Can a similar metric be derived for your case? For instance, ELBO surgery [1] rewrites the vanilla KL term (the KL between q(z_i|x) and p(z)) as the KL between the aggregate posterior and prior (KL[ q(z) || p(z) ]) + an extra mutual information term (also mentioned in the "Related work" section of the original WAE paper). In that case, if you use a WAE-GAN, the discriminator estimates KL[ q(z) || p(z) ], and you could use that to obtain a rough lower bound on log p(x) on the test set. In fact, that in some sense makes WAE-GAN slightly more appealing to use here than what you have presented. What are your thoughts on this?
- It's not clear to me whether using MoCA would be preferable from a more computationally 'economic' perspective than WAE-GAN. In the cases of both, hyperparameters are necessary, and GAN training has been made easier in recent times with the right 'tricks' (say, careful normalisation of the discriminator and using WGAN/JSGAN). Can something be said about the memory/computation time tradeoffs between using WAE-GAN and MoCA? For instance, if you're using contrastive learning, you either need a large batch size (for SimCLR) or a large queue (MoCA), whereas with GAN training you don't need this.
The main contribution is the nice theoretical connection between WAE training and contrastive learning. However, I am concerned with the statistical significance of some of the results.

Justification of rating: my main concern is in the statistical significance of the results in Table 1. I am open to raising my score if my concerns have been addressed.

References:
- [1]: Hoffman, M. D., & Johnson, M. J. (2016, December). Elbo surgery: yet another way to carve up the variational evidence lower bound. In Workshop in Advances in Approximate Bayesian Inference, NIPS (Vol. 1, p. 2).

---

### Official Review · AnonReviewer2 · 2020-10-28
**Review Summary for Momentum Contrastive Auto-Encoder**

**Rating:** 3
**Confidence:** 5

**Review:**

The paper is fairly easy to follow and does not have serious presentation problems. Especially, the discussion on the background (WAE) is clear and concise.

None the less, the two weaknesses are
1. The idea of having a hyperspherical latent space [1] is already been studied (and is a famous work), yet the author does not include any discussion with it. Even the author has discussed it, I can't see any difference between using the methodology presented in prior work and the submission.
2. There is nothing new I can learn from this submission. Precisely, I can't find the presented approach has any benefits over existing variational auto-encoder variants.

Suggestion:
1. I can't agree that momentum encoder makes contrastive learning "more computational efficient". It requires extra handing on the momentum. I think what the author likes to claim is that the momentum encoder leverages an asynchronous update and hence the effective batch size is larger.
2. In section 5.4, I can't agree with the statement from the author "for larger temperature, the estimated distribution becomes smoother and the entropy estimation becomes poor, which should result in a poor quality of generated samples." This statement is very vague and is incorrect.

Typo:
1. Theorem 1: should be when Z~P_Z not when g ~ P_Z

[1] Hyperspherical Variational Auto-Encoders, Davidson et al., UAI 2018.

---

### Official Review · AnonReviewer3 · 2020-10-28
**review of Momentum Contrastive Autoencoder**

**Rating:** 5
**Confidence:** 4

**Review:**

**Summary**
This paper considers training autoencoders with a hyperspherical latent space distribution. The encoder maps inputs to latent variables with a unit norm, and a contrastive loss with momentum is used to encourage the latent variables to be distributed uniformly over the surface of the unit-hypersphere. Generations of this autoencoder model are compared against wasserstein auto-encoders and several VAEs and evaluated with FID scores on CIFAR10 and CelebA. On CelebA the model is outperformed by both WAE-GAN (Wasserstein auto-encoders trained with a GAN objective to match the latent distributions) and a 2-stage VAE. On CIFAR10 the model outperforms related work, but only when using a different encoder/decoder architecture.
Samples, reconstructions and interpolations are also depicted for Cifar10, CelebA and CelebA-HQ. The influence of hyperparameter settings are studied through various ablation studies.

**Pros**
* The paper is well written.
* The proposed idea is clearly presented and well motivated.
* Hyperparameter settings and their influence on performance are extensively studied.

**Cons**
* The claims on performance gains on both Cifar10 and CelebA are overstated. The abstract states that state of the art FID scores on CIFAR10 (54.36) are obtained, which is actually far from the current state of the art on CIFAR10 (which is around 3.17, achieved by [1]). A more representative claim would be one that states that the proposed method outperforms related methods on CIFAR10 when making some architecture changes. On CelebA the statement is that the model performs competitively with respect to related work, even though related work such as the 2-stage VAE has an FID score of about 10 points lower. I understand that compute resources are not equally distributed among researchers and that lots of resources are required to scale to massive models in order to reach state of the art performance. Comparisons to related work on smaller models with comparable architectures is also a perfectly valid way to demonstrate the workings of a new algorithm/model. However, it would be more transparent if the results would then be framed in this context.
* In table 1 with the FID scores, it seems like most of the performance gains of the proposed model are actually obtained by changing the architecture of the encoder/decoder that is used in related work to a ResNet-18-based architecture with much fewer parameters. Importantly, with an architecture comparable to related work the proposed method does not improve upon related work on CIFAR10. To disentangle the performance gains due to the architecture change and the proposed method, at least some of the related methods should be evaluated with this different ResNet-18-based architecture. With the current results table it is hard to conclude if there are any benefits to using this contrastive auto-encoder framework compared to the other related methods.
* In section 5.1 the interplay of the latent dimensionality ($d$) and the hyperparameter ($\lambda$) that controls the contribution of the latent contrastive loss compared to the reconstruction loss is discussed. The observation is made that a larger dimensionality $d$ requires a higher value of $\lambda$ in order to obtain better FID scores (Fig 3). A hypothesis for this is given along the lines of there being holes in the latent space when the intrinsic data dimensionality is lower than that of the latent dimensionality, and that larger $\lambda$ will make sure to distribute the latent representations of datapoints more uniformly. However, wouldn’t the following argument also explain the need for a larger $\lambda$ for a higher $d$:  The probability of two random unit-norm encodings being orthogonal to each other increases with the dimensionality. This can lead to a larger probability of obtaining inner products with small magnitudes in the exponent of eq 6 and hence a contrastive loss that is closer to zero. Therefore, to rescale the contribution of the contrastive loss with respect to the reconstruction loss (which is independent of latent dimensionality) one needs to increase the $\lambda$. Could this also be the reason?


**Minor comments/questions**
* There are various other related works on auto-encoders with hyperspherical latent spaces that are not mentioned, such as [2, 3]. Please discuss more related work with hyperspherical priors.
* Please add a citation to [4] when referring to VAE’s in the second paragraph of the related work.
* The sample quality of VAEs has improved rapidly recently, see for instance [5]. The related work section on the sample quality of VAEs could reflect this a bit more.
* Typo in paragraph starting with Ghosh et al. in related work: “the further…” → “they further…”
* In theorem 1 in the end of the second line, it states $g\sim P_Z$. Should this be $Z\sim P_Z$?
* A factor of $1/K$ seems to be missing in the equation for $-L^{MC}_{neg}$ in the paragraph below eq. 4.

[1] Ho et al., denoising diffusion probabilistic models, https://arxiv.org/abs/2006.11239
[2] Patrini et al., Sinkhorn autoencoders, https://arxiv.org/abs/1810.01118
[3] Davidson et al., Hyperspherical Variational Auto-encoders, https://arxiv.org/abs/1804.00891
[4] Rezende et al., Stochastic backpropagation and approximate inference in deep generative models, https://arxiv.org/abs/1401.4082
[5] Vahdat & Kautz, NVAE: A deep hierarchical variational autoencoder, https://arxiv.org/abs/2007.03898

---

### Decision · Program_Chairs · 2021-01-07
**Final Decision**

**Decision:**

Reject

**Comment:**

This paper presents an interesting approach for training generative autoencoders with a latent space that lies on a hyperspherical subspace. However, the reviewers have raised concerns regarding the similarity of this work with several prior works and have questioned the experimental setup. Without the authors' response, we cannot situate this paper among the prior work properly. Thus, I recommend Reject at this point.